# Different Species of Bats: Genomics, Transcriptome, and Immune Repertoire

**DOI:** 10.3390/cimb47040252

**Published:** 2025-04-07

**Authors:** Huifang Wang, Hao Zhou, Xinsheng Yao

**Affiliations:** 1Department of Immunology, Center of Immunomolecular Engineering, Innovation & Practice Base for Graduate Students Education, Zunyi Medical University, Zunyi 563000, China; 18239577551@163.com; 2Department of Genome Informatics, Research Institute for Microbial Diseases, Osaka University, Suita 5650801, Japan; zhouhao@biken.osaka-u.ac.jp; 3Department of Systems Immunology, Immunology Frontiers Research Center, Osaka University, Suita 5650801, Japan

**Keywords:** Chiroptera, immunology, genomics, transcriptome, immune repertoire

## Abstract

Bats are the only mammals with the ability to fly and are the second largest order after rodents, with 20 families and 1213 species (over 3000 subspecies) and are widely distributed in regions around the world except for Antarctica. What makes bats unique are their biological traits: a tolerance to zoonotic infections without getting clinical symptoms, long lifespans, a low incidence of tumors, and a high metabolism. As a result, they are receiving increasing attention in the field of life sciences, particularly in medical research. The rapid advancements in sequencing technology have made it feasible to comprehensively analyze the diverse biological characteristics of bats. This review comprehensively discusses the following: (1) The assembly and annotation overview of 77 assemblies from 54 species across 11 families and the transcriptome sequencing overview of 42 species from 7 families, focused on a comparative analysis of genomic architecture, sensory adaptations (auditory, visual, and olfactory), and immune functions. Key findings encompass marked interspecies divergence in genome size, lineage-specific expansions/contractions of immune-related gene families (APOBEC, IFN, and PYHIN), and sensory gene adaptations linked to ecological niches. Notably, echolocating bats exhibited convergent evolution in auditory genes (SLC26A5 and FOXP2), while fruit-eating bats displayed a degeneration of vision-associated genes (RHO), reflecting trade-offs between sensory specialization and ecological demands. (2) The annotation of the V (variable), D (diversity), J (joining), and C (constant) gene families in the TR and IG loci of 12 species from five families, with a focus on a comparative analysis of the differences in TR and IG genes and CDR3 repertoires between different bats and between bats and other mammals, provides us with a deeper understanding of the development and function of the immune system in organisms. Integrated genomic, transcriptomic, and immune repertoire analyses reveal that bats employ distinct antiviral strategies, primarily mediated by enhanced immune tolerance and suppressed inflammatory responses. This review provides foundational information, collaboration directions, and new perspectives for various laboratories conducting basic and applied research on the vast array of bat biology.

## 1. Introduction

In 1991, the National Institutes of Health launched the human genome sequencing program, completing its initial draft in 2001, which propelled the development of genomic research in life sciences. In recent years, sequencing technologies have undergone transformative progress, transitioning from single-fragment Sanger sequencing to next-generation sequencing (NGS) platforms (e.g., Illumina short-read systems) and, more recently to long-read sequencing technologies (e.g., PacBio SMRT and Oxford Nanopore). These technological advancements have been applied across diverse organisms and multiple omics [1,2,3]. Bats, capable of hosting numerous viruses without exhibiting overt disease symptoms, have drawn attention to their immune systems. Genomic studies provide critical tools and insights for elucidating bat tolerance mechanisms [4]. However, due to the vast diversity of Chiroptera with 20 families, 1213 species, and over 3000 subspecies, it is a very difficult task to comprehensively sequence the genome of bats [5]. Currently, several projects, such as BAT1K [6], VGP [7], CCGP [8], Zoonomia [9], and DNA Zoo (https://www.dnazoo.org/) are dedicated to unraveling the mysteries of bats through genomic analysis. In 2011, *Myotis lucifugus* was the first bat genome to be entirely sequenced using the Sanger method [10]. To date, the NCBI database includes 77 genome assemblies from 54 bat species belonging to 11 families (Table 1).

As genome research progresses, determining the role of individual genes emerges as a formidable challenge. To address this, DNA microarray technology enables researchers to simultaneously observe the expression of thousands of genes, accelerating the understanding of gene expression regulation [11]. Building on this foundation, second-generation high-throughput sequencing technologies such as RNA-Seq have further deepened functional genomics insights through the precise characterization of gene activity and cellular dynamics [12,13]. However, these technologies have limitations in read length and accuracy. Third-generation sequencing technologies (such as PacBio’s SMRT and Oxford Nanopore’s nanopore sequencing) have overcome these limitations, providing broader and more precise transcriptomic data [14], and have also propelled the development of single-cell transcriptomics [15]. To date, transcriptome sequencing has been carried out for 42 bat species spanning seven families (Table 2). These data not only provide a rich resource for studying the immune adaptation, metabolism, hibernation, and other characteristics of bats but also offer important clues for understanding how bats tolerate viral infections.

The adaptive immune receptor repertoire (AIRR) has been widely used for assessing organisms’ healthy or disease status. However, comprehensive germline gene information is necessary for the downstream analysis. Since 2010, Baker et al. had been analyzing the heavy chain variable genes of the black flying fox, *Pteropus alecto* [16]. Subsequently, our group conducted a comprehensive analysis and comparison of the TCR and BCR germline genes across three bat species, and we also further analyzed the bat T/B repertoire using high-throughput sequencing [17,18,19]. Currently, TCR and BCR gene annotations are underway in various laboratories for 12 bat species belonging to five families (Table 3).

In this review, we have summarized the genome, transcriptome, and immune repertoire sequencing data of bats (Figure 1), underscoring the significance of these datasets in elucidating the biological processes of these species (Figure 2). While innate immune components such as MHC and TLRs have been previously discussed by other scholars [20,21], our analysis builds upon their work by incorporating the latest findings in these areas. Our primary emphasis is on the synthesis of data pertaining to bats’ adaptive immune systems, including the annotation of TR (T cell receptor) and IG (immunoglobulin) genes across various bat populations and the analysis of CDR3 (Complementarity determining region 3) repertoires from high-throughput sequencing data, which shed light on the VDJ recombination, diversity, and characteristics of the CDR3 in adaptive T and B cells.

**Table 2 cimb-47-00252-t002:** A transcriptome analysis of different species of bats in different studies.

Suborder	Family	Species	Accession Number	Samples	Reference
Microchiroptera	Phyllostomidae	*Artibeus jamaicensis*	PRJNA490553	liver	[22]
			PRJNA305413	kidney, liver, and spleen	[23]
			PRJNA172870	lung, kidney, and spleen	[13]
			PRJNA929543	wings	[24]
		*Carollia brevicauda*	PRJNA563501	vomeronasal	[25]
		*Carollia perspicillata*	PRJNA563501	vomeronasal	[25]
			PRJNA291081	limbs	[26]
		*Desmodus rotundus*	PRJNA139591	ganglia	[27]
			PRJNA563501	vomeronasal	[25]
			PRJNA531931	principal and accessory submaxillary glands	[28]
			PRJNA451361	salivary glands	[29]
			PRJNA174817	principal submaxillary gland	[30]
		*Phyllostomus hastatus*	PRJNA563501	vomeronasal	[25]
		*Phyllostomus discolor*	PRJNA572574	muscle	[31]
			PRJNA291690	brain	[32]
		*Erophylla sezekorni*	PRJNA929543	wings	[24]
	Vespertilionidae	*Eptesicus fuscus*	PRJNA300284	liver and spleen	[22]
			PRJNA591156	wings	[33]
		*Myotis myotis*	PRJNA949638	olfactory nerve-derived cell	[34]
			PRJNA591156	wings	[33]
			PRJNA572574	muscle	[31]
			PRJNA503704	blood	[35]
			PRJNA277738	wing skin	[36]
			PRJNA267654	blood	[37]
		*Myotis lucifugus*	PRJNA300284	liver and spleen	[22]
			PRJNA826885	primary embryonic fibroblast cells	[38]
			PRJNA591156	wings	[33]
			PRJNA451361	salivary glands	[29]
			PRJNA277738	wing skin	[36]
			PRJNA246229	wing	[33]
			PRJNA218522	submandibular gland	[39]
		*Myotis brandtii*	PRJNA300284	liver and spleen	[22]
			GSE42297	liver, kidney, and brain	[40]
		*Myotis ricketti*	PRJNA290538	brain	[41]
			PRJNA640936	cochlear	[42]
			PRJNA198831	inner ear	[43]
			PRJNA142913	brain and adipose	[44]
		*Myotis pilosus*	PRJNA902548	cochlea, brain, muscle, liver, and heart	[45]
			PRJNA290538	brain	[41]
		*Myotis velifer*	PRJNA625724	embryonic fibroblasts	[46]
		*Myotis daubentonii*	PRJNA496612	kidney cell	[47]
		*Myotis laniger*	PRJNA175507	brain	-
			PRJNA255191	-	-
		*Myotis davidii*	PRJNA300284	liver and spleen	[48]
			PRJNA172130	-	[49]
		*Murina leucogaster*	PRJNA290538	brain	[41]
		*Miniopterus schreibersii*	PRJNA218524	forelimb	[50]
		*Miniopterus natalensis*	PRJNA270639	forelimbs	[51]
		*Miniopterus fuliginosus*	PRJNA175507	brain	-
		*Scotophilus kuhlii*	PRJNA290538	brain	[41]
		*Pipistrellus kuhlii*	PRJNA572574	muscle	[31]
	Megadermatidae	*Megaderma lyra*	PRJNA290538	brain	[41]
	Molossidae	*Tadarida brasiliensis*	PRJNA184055	liver, kidney, and brain	[52]
		*Tadarida teniotis*	PRJNA290538	brain	[41]
		*Molossus molossus*	PRJNA572574	muscle	[31]
	Rhinolophidae	*Rhinolophus ferrumequinum*	PRJNA300284	liver and spleen	[48]
			PRJNA572574	muscle	[31]
		*Rhinopoma hardwickei*	PRJNA290538	brain	[41]
		*Rhinolophus pusillus*	PRJNA290538	brain	[41]
			PRJNA972471	liver	[53]
		*Rhinolophus affinis*	PRJNA640936	cochlear	[42]
			PRJNA644044	cochleae	[54]
	Emballonuridae	*Taphozous melanopogon*	PRJNA290538	brain	[41]
	Hipposideridae	*Aselliscus stoliczkanus*	PRJNA290538	brain	[41]
		*Hipposideros pratti*	PRJNA290538	brain	[41]
		*Hipposideros armiger*	PRJNA300284	liver and spleen	[48]
			PRJNA900132	cochlea, brain, heart, liver, and muscle	[55]
Megachiroptera	Pteropodidae	*Cynopterus sphinx*	PRJNA290538	brain	[41]
			PRJNA198831	inner ear	[43]
		*Eonycteris spelaea*	PRJNA847556	lung	[56]
			PRJNA427241	lung and kidney	[14]
			PRJNA530519	spleen	[57]
		*Pteropus vampyrus*	PRJNA300284	liver and spleen	[48]
		*Pteropus alecto*	PRJNA397372	IFN-stimulated cells	[58]
			PRJNA531095	spleen	[57]
			PRJNA172130	-	[49]
		*Pteronotus quadridens*	PRJNA929543	embryo	[24]
		*Rousettus aegyptiacus*	PRJNA300284	liver and spleen	[48]
			PRJNA902735	PBMC	[59]
			PRJNA762341	PBMC	[60]
			PRJNA640284	blood, kidney, liver, and spleen	[61]
			PRJNA572574	muscle	[31]
		*Rousettus leschenaultii*	PRJNA290538	brain	[41]

**Table 3 cimb-47-00252-t003:** The TCR/BCR gene annotation of different bat species in a different study.

	Family	Species	Chain	V Genes	D Genes	J Genes	C Genes	Reference
TR LOCUS	Phyllostominae	*Phyllostomus discolor*	TRB	100	2	13	2	[17]
	Rhinolophinae	*Rhinolophus ferrumequinum*	TRA	81	0	60	0	[17]
			TRD	18	1	4	1	[17]
			TRB	29	2	15	2	[17]
			TRG	14	0	6	2	[17]
	Vespertilionidae	*Pipistrellus pipistrellus*	TRB	45	2	13	2	[18]
	Multiple (10)	*Multiple* (14)	ALL TR locus	-	-	-	72	[62]
IG LOCUS	Phyllostominae	*Phyllostomus discolor*	IGH	81	16	7	IGHM, IGHE, IGHG, IGHA	[19]
	Rhinolophinae	*Rhinolophus ferrumequinum*	IGH	41	4	6	IGHM, IGHE, IGHG, IGHA	[19]
	Vespertilionidae	*Pipistrellus pipistrellus*	IGH	57	7	6	IGHM, IGHE, IGHG, IGHA	[19]
	Pteropodinae	*Rousettus aegyptiacus*	IGH	58	8	3	IGHM, IGHE, IGHG, IGHA	[63]
	Molossidae	*Eumops perotis* (Only Functional)	IGH	10	8	3	5	[64]
	Vespertilionidae	*Myotis lucifugus*	IGH	>250	-	-	-	[65]

## 2. Genomics of Different Bat Species

### 2.1. Genetic Basis of Bat Immune Adaptations

Gene expansion and contraction within the bat genome suggest that bats may have evolved distinct strategies to respond to antigen invasions. Emerging evidence suggests that bat immune systems balance viral coexistence and host survival through dual strategies: immune tolerance and immune resistance. Notably, some bats maintain rapid antiviral defenses via constitutively active innate immunity. For example, comparative studies in *Rousettus aegyptiacus* and *Myotis myotis* show both species depend on interferon signaling, yet diverge in pathway utilization. *Rousettus aegyptiacus* prioritizes TLR3 activation, while *Myotis myotis* relies on the RIG-I pathway, highlighting evolutionary flexibility in antiviral tactics [66]. Levinger R et al. used single-cell and spatial transcriptome technology to reveal the unique distribution of immune cells (such as the high proportion of regulatory T cells) in bat skin and respiratory barrier tissues and the low expression of virus receptors (such as ACE2) in epithelial cells, limiting virus invasion [67]. Other studies have shown that there are differences in interferon type I among different species of bats, such as Pavlovich et al. [68] discovered positive selection in *Rousettus aegyptiacus* for interferon-stimulated genes ISG15, type I IFN receptor subunit IFNAR1, and interferon response negative regulator SIKE1 in *Rousettus aegyptiacus*. The IFN-ω gene has expanded extensively to nearly 20 copies in this species, a contrast to the expansion observed in *Pteropus alecto* but is consistent with *Pteropus vampyrus*.

In the tolerance mechanism, bats can achieve an asymptomatic carrying of highly pathogenic viruses (such as the Ebola virus and Marburg virus) by regulating inflammatory response and metabolic adaptation, including IRF1, which inhibits excessive inflammation by up-regulating immune checkpoint molecules (such as PD-L1) [57], and inhibits the expression of proinflammatory factors (IL-6, TNF-α, IL-1β, and IL-8) to alleviate tissue damage [61,69]. The high-frequency expression of IGHV1 family genes in B cells is used to produce broad-spectrum antibodies, and low-inflammatory antibody subtypes such as IgG4 are relied on to reduce immune activation [63], and the up-regulation of the T cell immune checkpoint (PD-1/CTLA-4) inhibits overreaction and enhances antioxidant defense (such as the high expression of SOD2 and the deletion of PYHIN family sites) to alleviate virus-induced oxidative stress [49,70,71], so as to maintain the host–pathogen balance under the persistent existence of a virus. In addition, Vicente-Santos A et al. found that the serum of *Desmodus rotundus* highly expressed complement inhibitory proteins (such as CFH) and anti-inflammatory factors (such as IL-10), which supported a wide tolerance to many pathogens [72].

Genes mediating a variety of cellular functions in bats have experienced different levels of evolutionary pressure. The genomic studies of *Cynopterus brachyotis* and *Phyllostomus hastatus* revealed notable reductions in the genes encoding natural killer (NK) cell receptors. Although the KLRK1 genes were retained, the ligands for this receptor, along with the RAET1 and H60 gene subfamilies that activate NKG2D, were conspicuously missing [71,73]. By comparative analysis of the genomes of Rhinolophidae bats, Hipposideridae bats, and other mammals, it has been found that there is a deletion of Cys78 in the ISG15 gene of bats, which is not present in humans. These differences result in a reduced function of ISG15 in bats regarding viral recognition and signal transduction. These findings suggest that bats may maintain immune balance through other compensatory mechanisms, thereby carrying viruses without showing obvious symptoms of disease [74]. Tian et al. found the immune genes of *Cynopterus sphinx* have undergone rapid evolution and possess unique variations, including the loss of the NLRP1 gene, duplication of the PGLYRP1 and C5AR2 genes, and changes in amino acids within the MyD88 gene. This suggested that comparing bats across different species, as well as contrasting the immune genes of bats with those of other mammals, is an important direction for deciphering the evolutionary pressures on these species [75]. The APOBEC gene family has been found to produce expansions in the genomes of several bats, including a second APOBEC3 locus in Myotis, which may contribute to their viral tolerance [73].

### 2.2. Sensory Gene Adaptations in Bats: Hearing, Sight, and Smell

As echolocating mammals primarily dependent on high-frequency sound perception for nocturnal navigation and prey detection, bats exhibit evolutionary adaptations in sensory systems. For instance, Parker J et al. [76] studied seven auditory (SLC26A5, Tmc1, Kcnq4 (Kqt-4), Pjvk (Dfnb9), otoferlin, Pcdh15, and Cdh23) genes that showed convergence or adaptation in echolocating mammals. They found that this convergence phenomenon is observed in both echolocating bats (*Rhinolophus ferrumequinum* and *Megaderma lyra*) and non-echolocating bats (*Eidolon helvum* and *Pteronotus parnellii*). This pattern extends to inner-ear gene loci (SLC45A2, RGS7BP, TRPV5, and NOX3), where parallel functional adaptations emerge in bats and dolphins facing analogous acoustic challenges [40]. Investigations in *Myotis davidii* and *Pteropus alecto* further corroborate echolocation-related selection pressures, with the FOXP2 gene displaying elevated nonsynonymous-to-synonymous substitution ratios (dN/dS) relative to non-echolocating species. Additionally, exon 3 of the FOXP2 gene exhibited more variation than the typical mammalian sequence, suggesting unique adaptations in bat species [49].

While vision is less critical for bats than hearing and smell, visual systems demonstrate contrasting evolutionary trajectories. Parker et al. [76] found a contraction of vision-related genes (such as Itgal) in *Rhinolophus ferrumequinum*, which is associated with the sophisticated echolocation and low-light conditions found in the Rhinolophidae bat family. Moreover, Dong et al. [77] found that only 3 out of 200 vision genes (RHO, CRX, and SAG) in *Myotis lucifugus* had degenerated into pseudogenes, in contrast to 12 (such as OPN1SW, PDE6B, and GUCY2D) in *Hipposideros armiger* and *Rhinolophus sinicus*. The evolution of vision, hearing, and smell in bats has led to varying degrees of divergence among different species. To understand these variations at the gene level, more bat species need to have their entire genomes sequenced.

### 2.3. Flight, Metabolism, and Lifespan in Bats

There is an evolutionary correlation mechanism between flight, metabolism, and the lifespan of bats. Comparative genomics of *Myotis davidii* and *Pteropus alecto* uncovered positive selection in DNA repair pathways, notably LIG4 under positive selection in *Pteropus alecto* [49]. Parallel studies in *Ia io* identified positively selected genes (DDB1, PHF1, and ERCC6) associated with hypoxic tolerance and oxidative damage repair, equipping this species to withstand metabolic extremes during high-altitude predation [78].

Intriguingly, bats exhibit adaptive co-evolution between dietary ecology and energy metabolism strategies: fruit bats exhibit suppressed tyrosine aminotransferase activity (implying protein synthesis-dominated energy production) [79]. Whereas the insectivorous *Lasiurus cinereus* displays accelerated evolution in 17 lipid/energy metabolism genes and 15 mitochondrial regulators, including those influencing the sensitivity to metabolic hormones and encoding lipid-metabolizing enzymes and proteins [80]. Gong et al. [78] identified evolutionary adaptations in *Ia io*, where genes related to iron metabolism evolved to help it cope with high iron ions and fats from its bird digestion, preventing iron toxicity.

Bats exhibit long lifespans, potentially due to their high tolerance against oxidative stress. In cross-species comparison, *Myotis myotis* showed a higher expression level of DNA repair gene (CREBBP) network than *Molossus molossus*, and the transcription activation of autophagy-related pathways (PIK3R1) was more obvious, which may contribute to enhanced DNA repair and autophagy activities [81]. Combined with transcriptomes in other long-lived species, this suggests that enhanced DNA repairing and autophagic activity may be generalized mechanisms through which mammals achieve longevity [82]. In contrast to *Pteronotus mesoamericanus*, a positive selection of genes related to DNA damage repair and 33 tumor suppressor genes were also seen in *Artibeus jamaicensis*. This may provide insights into the low incidence of cancer and the long lifespan in bats [83]. Chattopadhyay et al. [73] found telomeres in Myotis bats were observed to not shorten with age, and genes ATM and SETX protect telomeres from oxidative stress, potentially contributing to bats’ longevity. The aging genes Growth Hormone Receptor (GHR) and Insulin-Like Growth Factor 1 Receptor (IGF1R) are also linked to a potential association with the extended lifespan of bats, including transmembrane domain deletions in Vespertilionoidea and cysteine-deficient mitochondrial isoforms in *Myotis brandtii* [84]. Investigation into the relationship between flight, metabolism, and lifespan in bats has so far only involved a preliminary genetic analysis of physiological adaptations to energy management strategies and oxidative stress. More detailed genetic annotations and analyzes are needed to fully understand the biological characteristics of bats.

## 3. Transcriptomes of Different Bats Species

### 3.1. Immune Transcriptomes Across Bat Species

Recent advances in transcriptomic analyses have revealed multifaceted antiviral adaptations across bat species, providing insights into their unique viral tolerance. Moreno-Santillán et al. [22] identified species-specific protein networks (GAG, POL, and ENV) in hepatic tissues of five tropical bats, suggesting co-opted endogenous pathways for viral replication regulation. Researchers also have identified specific expression patterns of various virus receptor genes across different tissues in bats, and some have found genes encoding respiratory virus entry factors to be uniquely expressed in bats through comparative transcriptome analysis with other species [15,45,85]. Notably, transcriptomes of *Rousettus aegyptiacus* uncovered a constitutive expression of antiviral effectors (IFN-γ and IFN-α) and pattern recognition receptors, yet strikingly lacked baseline IFN-β expression, consistent with other mammals in which IFNB is expressed only after stimulation [48]. What is more, no evidence of constitutive IFN expression was found in *Rousettus aegyptiacus* compared with *Pteropus alecto*, suggesting different mechanisms for controlling viruses in these two bat species [68]. Single-cell sequencing technology enables the in-depth exploration of the gene expression profiles within specific cell populations in different organisms. Gamage et al. [56] observed that different genes were expressed by different cell types in lung tissue following PRV3M infection in *Eonycteris spelaea*. Friedrichs et al. [60] discovered that *Rousettus aegyptiacus* juvenile, subadult, and adult B and T cells have expression patterns that differ from those of other mammals and that B-cell clusters have distinct patterns of a high expression of CD20 and CD19, whereas T cells express TCR–CD3 complex components. With the development of single-cell sequencing technology, an in-depth exploration of the gene expression profiles in different types of bat immune cells is now feasible, allowing for insights into their diversity and immune functionality.

### 3.2. Gene Expression in Bat Metabolism, Lifespan, and Hibernation

Bats are a group of mammals characterized by remarkable metabolic adaptations and long lifespans. They are able to survive under extreme conditions, including prolonged hibernation and the ability to fly with high energy consumption. Zhang et al. [49] reported changes in the RNASE4 digesting enzyme in *Myotis davidii* and *Pteropus alecto*, which may help bats more effectively process complex carbohydrates in insects while also providing immune protection during viral infections. Complementing this, salivary gland transcriptomes of *Myotis lucifugus* reveal seven autocrine lipases that repurpose dietary lipids into flight metabolism, suggesting dietary adaptations in bat salivary glands [39]. Building on these findings, comparative liver transcriptomics across five Neotropical bats (*Artibeus jamaicensis*, *Mormoops megalophylla*, *Myotis keaysi*, *Nyctinomops laticaudatus*, and *Peropteryx macrotis*) uncover dietary-driven expression divergence: Frugivorous *Artibeus jamaicensis* upregulates ALDOB (a fructose-metabolizing enzyme), whereas insectivores (*Mormoops megalophylla*, *Myotis keaysi*, etc.) predominantly express serum albumin for nitrogen scavenging [22]. Notably, the conserved overexpression of APOE across all species implicates lipid transport in dual metabolic–immunological regulation.

Hibernation physiology in *Rhinolophus ferrumequinum* exposes a dynamic gene regulation tied to energy conservation, such as PDK4 [12], PRL-2 [86], and EST [87] in different tissues of *Rhinolophus ferrumequinum*. Meanwhile, some groups analyses found that genes related to energy metabolism, the cellular stress response, oxidative stress, and cytoskeletal organization were overexpressed during bat hibernation [88,89]. The low expression of the nine genes (PFKP, PFKM, PFKL, ALDOA, ALDOC, ENO1, ENO2, GPI, and PKM) involved in glycolysis in hibernating bats supports the idea that glucose conservation is a common strategy in mammals during hibernation. Investigations in *Myotis brandtii* link lipid-to-carbohydrate metabolic shifts to starvation adaptation genes (CPT1A and ANGPTL4) and coagulation regulators [40]. Additionally, a comparative analysis of gene expression between GHR mice and *Myotis brandtii*’s liver revealed elevated expression levels of multiple insulin signaling-related genes in both, along with potential associations between changes in the GHR (growth hormone receptor) gene expression and the extended lifespan observed in bats.

### 3.3. Audiovisual-Related Genes in Bats

The auditory system of bats plays a crucial role in their nocturnal hunting. Researchers are investigating the cochlear transcriptomes of various bat species to uncover variations in gene expression that play a role in perceiving sounds across different frequencies and in sound localization. In comparative analyses of three phylogenetically distinct bats, a striking upregulation dominance in the cochlear gene expression of constant-frequency (CF) specialists (*Rousettus leschenaultii*) was revealed. Zhao et al. [90] identified 18 hearing-associated unigenes in *Rhinolophus ferrumequinum* through multi-cochlear RNA sequencing, with functional enrichment in synaptic binding (SV2B-like), ion transport (AQP3), and immunoregulatory pathways (IGSF3-like). In *Myotis pilosus*, Wang et al. [45] discovered ultrasonic hearing-associated transcripts linked to cochlear membrane mechanics, while Sun et al. [54] correlated FBXL15 expression levels with an echolocation frequency divergence across three Rhinolophus affinis subspecies (*Rhinolophus affinis healayanus*, *Rhinolophus affinis hainanus*, and *Rhinolophus affinis himalayanus*). Seim et al. [40] identified amino acid alterations in auditory-related genes and a gene associated with visual function (CNGB3) in *Myotis brandtii*. They also pinpointed new candidate genes (TRPV5 and NOX3) suggestive of functional adaptations in bats. Bat transcriptome analyses have uncovered gene expression differences linked to visual and auditory functions, as well as neural pathways. These discoveries establish a foundation for advancing our understanding of the sensory abilities of bats and contribute to unraveling the unique adaptive evolution of these specialized mammals in vision and hearing.

### 3.4. A Transcript Analysis of Other Genes in Bats

With the rapid advancement of transcriptomic technologies, we are provided with a window into the intricate patterns of gene expression within cells. Huang Z et al. [37] identified 37 miRNAs in *Myotis myotis* with conserved homology to humans and livestock, and three upregulated candidates demonstrate tumor-suppressive potential across human malignancies, whereas one downregulated species accelerates breast/pancreatic oncogenesis. Wang et al. [45] conducted a transcriptomic analysis of the cochlea, brain, muscle, liver, and heart tissues of *Myotis pilosus*, revealing 1528 selective splicing events across six different types. Wen et al. [14] also found varied alternative splicing in *Eonycteris spelaea*. These findings can also be utilized to compare bat genomes and identify genetic variations that may influence bat evolution. Jebb et al. [31] sequenced and annotated the genomes of six species of bats and found MiR-337-33p showed a broad expression and involvement in developmental, rhythmic, synaptic, and behavioral gene pathways unique to bats. These findings contribute to understanding the complexity, diversity, and functional aspects of the bat transcriptome. From the advancements in gene chip technology to modern RNA sequencing, transcriptomics is rapidly on the rise. With the development of full-length transcriptome sequencing and the emergence of single-cell transcriptomics, we can gain a more comprehensive understanding of transcriptomes within cells or the diversity of cell types.

## 4. The Immune Repertoire of Different Bat Species

### 4.1. Bat TCR Gene and Repertoire

The adaptive immune system in bats has been studied restrictively because of the lack of specific reagents and tools. Research indicates that the CD4+ and CD8+ T cell proportions of bat lymphocytes are varied in different tissues [91]. In addition to that, bat immune cell proportions varied in different stages [60]. Otherwise, bats’ innate immune cells and molecules show significant differences from those of human and mouse, suggesting that they may play an important role in viral tolerance [20,92]. Our group was the first to annotate the TR and IG loci of different bats, and we found differences in the composition of germline VDJC genes between bats and other mammals. We also discovered a significant heterogeneity in the TCR β CDR3 repertoire and IGH CDR3 repertoire of bats compared to human and mouse [17,18,19], suggesting that the adaptive immune responses of T and B cells in bats may also be an important mechanism in the viral tolerance of bats.

The immune repertoire (IR), used to evaluate the individual healthy and disease situations, has been a common weapon [93]. Mammalian T cells are mainly divided into two types: αβ and γδ. Each of the chains is composed of germline Variable (V), Diversity (D), Joining (J), and Constant (C) genes, while the α and γ chains do not contain D genes [94,95]. In fact, it is feasible to identify TCR/BCR-related information within the RNA-seq data. In 2012, the transcriptomes of immune tissues and cells from *Pteropus alecto* were analyzed, revealing a large number of TCRα- and TCRβ-related contigs and a small number of TCRγ- and TCRδ-related contigs [96]. In 2015, Lee and others found sequences of TCR α and TCR β chains in the transcriptome data of 11 tissues of *Rousettus aegyptiacus*, but did not find TCRδ and TCRγ [48]. These results suggest that bats likely possess αβ T cells as their predominant T cell type. However, this is not the whole story of the adaptive immune receptor. An existing problem is how to perform raw high-throughput sequencing data analysis, which needs complete germline gene annotation as the reference library. Searching for TCR/BCR germline genes from whole genomes in the early stage was extremely difficult due to the lack of high-quality genomes and specific tools. Since 2020, multiple projects dedicated to sharing high-quality bat genomes have been initiated, including VGP and Bat1K [6,7]. This project enables the possibility of conducting a comparative analysis of TCR/BCR loci and germline genes in different bat species. In a 2019 study, the TRC gene of different chains was researched and annotated from 14 bat species. They investigated the influence of ecological factors on the evolution of mammalian TRC genes, revealing habitat as a major driver shaping TRC repertoires and suggesting mechanisms of adaptation to ecological niches [62]. In 2021, for the first time, our group thoroughly annotated four TCR loci of *Rhinolophus ferrumequinum* by using three annotation methods which include the following: (1) IMGT-LIGMotif, a specialized tool for identifying germline genes, (2) RSSite, which reverse-engineers the VDJ gene by detecting the 12/23 RSS, and (3) homology gene mapping conducted using Geneious Prime software (Version:2025.0.3) [17]. In terms of locus organization, *Rhinolophus ferrumequinum* does not change a lot compared to common mammals. The TRD locus is embedded within the TRA locus, and it remains undetermined whether the TRA/D locus can share a portion of the V genes with the TRD locus, owing to the absence of experimental validation. The upstream region of the Greater Horseshoe bat’s TRB gene locus contains multiple V genes, while the downstream region consists of two D–J–C clusters. It suggests that the evolution of TRB locus in bats is similar to that of humans and mice, rather than resembling the TRB loci in ruminants like pigs and cows, which have three D–J–C clusters [97]. However, the TRG locus is not the same story. Unlike in humans and mice, which contain only one V–J–C and four V–J–C clusters, respectively [98,99], the bat TRG locus contains two totally same V–J–C clusters.

Previous studies have reported varying degrees of the contraction or expansion of immune-related genes in different bat species. Further, to explore the TCR locus among different bat species, we compare the TRB gene locus of *Pipistrellus pipistrellus* and *Phyllostomus discolor* with the *Rhinolophus ferrumequinum.* Significantly, we find that the *Phyllostomus discolor* has a greater TRBV gene number (100) than the other two bats (30 and 45). This difference is considerable even if only counting functional genes, which is likely to suggest a Chiroptera naïve T cell repertoire wide diversity. While somatic hypermutation (SHM) and class-switch recombination (CSR) have not been identified in conventional TCR diversity generation mechanisms, emerging evidence suggests that SHM may contribute to T cell receptor diversification under specific immunological pressures [100]. In the downstream repertoire analysis, some basic expression data, such as the length and amino acid usage of the CDR3 region, the high-frequency V, J genes, and V–J pairing, demonstrated that there were no large differences in the TCR CDR3 repertoire of *Rhinolophus affinis* between human and mouse.

### 4.2. Bat BCR Gene and Repertoire

B cells play a crucial role in the immune system as they are primarily responsible for producing antibodies to combat pathogens. The majority of investigations have concentrated on determining the antibody activity of bat B cells. However, the conclusion and results vary by different bat species and experiments. An earlier study found that IgA and IgM were maintained at high levels under low or high fungal infection doses, whereas a significant elevation of IgG was not detected until 8–9 weeks later [101]. Another study found the virus-specific IgG levels maintain 11 months following infection [102].

Studying the bat BCR gene and immune repertoire shed light on their genetic evolution and antigenic responses, with the mammalian BCR polypeptide chain being a tetramer comprising two heavy (IGH) and two light chains. In 2010, Baker et al. [16] found in *Pteropus alecto* and *Pteropus vampyrus* 23 VH segments from five families. Interestingly, the CDR3 region in *Pteropus alecto* lacked tyrosine but had a higher proportion of arginine and alanine. This led to the hypothesis that the lower tyrosine content and increased arginine and alanine may have resulted in the evolution of antibodies with a lower polyreactivity, forming only weak associations with antigens. Butler et al. [103] discovered the disulfide loops and conserved cysteine of Ig sequences of four bat species (*Myotis lucifugus*, *Eptiscus fuscus*, *Carollia perspicillata*, and *Cynopterus sphinx*), which showed a high homology to humans. In 2011, Bratsch et al. [65] discovered 236 VH3 sequences in *Myotis lucifugus*. Furthermore, a comparison with fetal piglets revealed a similarly low mutation frequency. The enormous V gene pool and low mutation suggested that bats are likely to rely more on combinatorial rearrangement and junctional diversity for forming the antibody repertoire.

Advanced sequencing technology offers us the opportunity to annotate the germline genes from the whole genome. Larson et al. [63] constructed and annotated the *Rousettus aegyptiacus* IGH locus, revealing expanded IGHV genes linked to the protective human antibodies against various viruses. Notably, they observed duplications in IgY, resulting in functional alterations within IgG and IgE subtypes, including the presence of multiple *IGHE*s in *Rousettus aegyptiacus*, a rarity among mammals. Furthermore, the species has distinct IgG subclasses, suggesting potential functional differences compared to humans and mice. IGDetective [64] is a tool that searches for the VDJ gene from far-ranging species genomes. The authors reported several bat species IGH genes; however, the limited number that they identified from four bat species (*Eumops perotis*, *Rhinolophus ferrumequinum*, *Phyllostomus discolor*, and *Pipistrellus pipistrellus*) genes does not support the bat diversity gene pool hypothesis. In our recent study group, we comprehensively annotated the IGH locus of *Rhinolophus ferrumequinum*, *Phyllostomus discolor*, and *Pipistrellus pipistrellus* by the same methods as mentioned above [19]. As expected, the VDJ gene number is varied within the three species. A consistent result with the previous study is that three bat species have four immunoglobulin genes but IgD is absent. In the *Rhinolophus affinis* BCR repertoire, the most abundant subclass is IgG, unlike in humans and mice where it is IgM. In terms of gene usage, while *IGHV1* and *IGHV4* are found at high frequencies in both bats and humans, *IGHV3* is exclusive to bats. *IGHJ4* has a high frequency in bats, humans, and mice [104].

## 5. Summary and Outlook

Analyzing the genome, transcriptome, and immune repertoire of different bats can clarify their mechanisms and tolerance strategies as hosts of various viruses, and provide new ideas and methods for research. The development of the sequencing and annotation of bat genomes has promoted the development of new methods for studying virus tolerance mechanisms and provided technical support for clarifying its mechanism of action, such as how recent genomic analyzes of bats have revealed diverse antiviral adaptations, including expansions of APOBEC3 loci, positive selection in interferon-stimulated genes (ISGs), and lineage-specific losses of pro-inflammatory receptors, suggesting that antiviral mechanisms in bats evolved through multiple independent pathways [74]. The in-depth analysis of bat transcriptome enables us to analyze the dynamic expression rules of its immune-related genes, such as the constitutively expressing interferon-α (IFN-α) in bats to maintain an antiviral state [68]. Notably, the multi-level gene expression regulation synergizes with the highly diverse BCR/TCR repertoire. The latter, generating broad-spectrum antigen-recognition receptors, enables a precise pathogen attack. These studies may represent the main strategy of bats to coexist with viruses.

The establishment of bat cell models provides an important tool for studying bats’ unique antiviral ability. For instance, Crameri G. et al. [105] established immortalized cell lines of *Pteropus alecto* in 2009 to examine their susceptibility to rabies and Hendra viruses, among others. Déjosez et al. [106] generated and prompted the differentiation of stem cells from *Rhinolophus ferrumequinum* and *Myotis myotis* to explore the differences with those of other mammals. Recent studies have further expanded our understanding of bat cell lines and their applications. Establishing primary and immortalized fibroblast cell lines from wing biopsy samples of the *Rousettus aegyptiacus* provides researchers with a cellular foundation for studying the unique evolutionary adaptations of bats, expanding the range of bat species that can be studied [107]. Another study showed that viral dynamics within bats can be accelerated, which may be related to the efficient suppression of viral replication by the bat immune system during the early stages of infection [108]. These findings highlight the potential of bat cell lines for studying pathogen genomics and unique immune responses. However, given the large number of bat species, collaboration among multiple laboratories is critical for advancing the analysis of bat biological characteristics in life sciences and medicine. It is necessary to carry out research in a coordinated and orderly manner and share data in a timely manner.

## Figures and Tables

**Figure 1 cimb-47-00252-f001:**
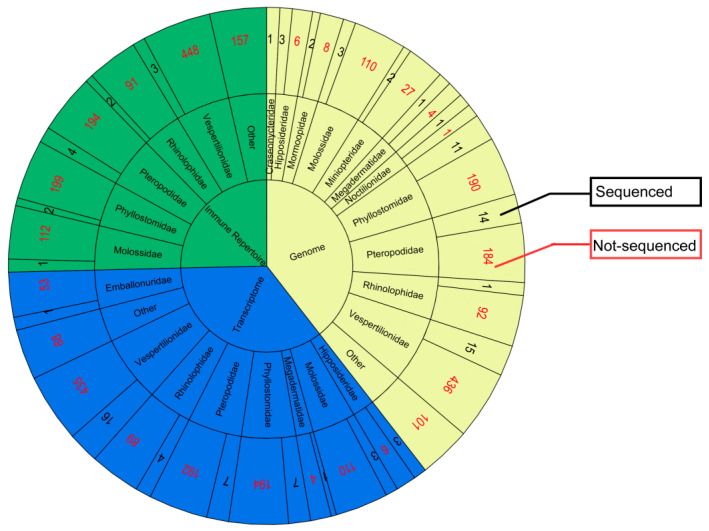
Sequencing profiles of the genomes, transcriptomes, and immune repertoire of different bat species. (The red font represents the number of non-sequenced bats, and the black font represents the number of sequenced bats).

**Figure 2 cimb-47-00252-f002:**
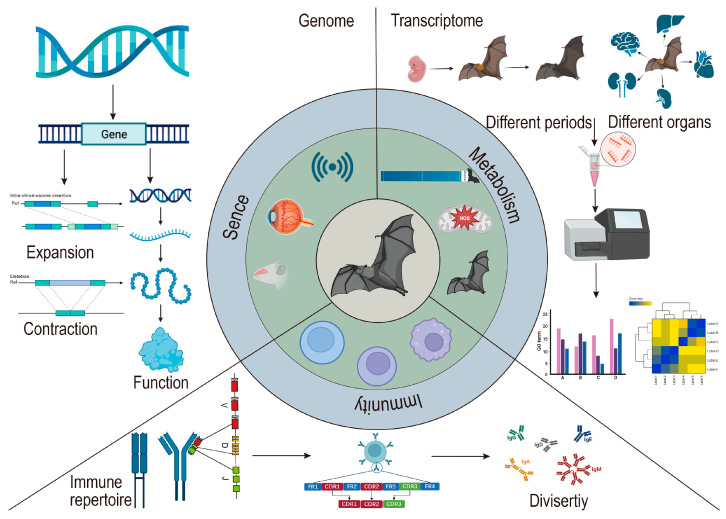
The bat genome, transcriptome, immune repertoire sequencing, annotation, and gene expression.

**Table 1 cimb-47-00252-t001:** Assembly information of bats in the public database.

Suborder	Family	Species/Assembly Accession Number	Total Length (Gb)	Unmapped Length (Mb)	Chromosome/ Scaffold/ Contig Count	Scaffold N50 (Kb)	Submitter
Microchiroptera	Vespertilionidae	*Corynorhinus townsendii* (Townsend’s big-eared bat)/GCA_026230055.1	2.1	0.02	390 Scaffold	174,690.16	UCLA
		*Antrozous pallidus* (pallid bat)/GCA_027563665.1	2.13	5.88	23 Chromosome	114,648.71	Bat1K
		*Ia io* (great evening bat)/GCA_025583905.1	2.1	0.06	2008 Scaffold	105,833.40	Northeast Normal University
		*Eptesicus fuscus* (big brown bat)/GCA_027574615.1	2.01	0.03	25 Chromosome	102,817.58	Bat1K
		*Aeorestes cinereus* (hoary bat)/GCA_011751065.1	2.15	42.36	2536 Scaffold	35,075.55	United States Geological Survey
		*Myotis lucifugus* (little brown bat)/GCA_000147115.1	2.03	68.16	11,654 Scaffold	4293.31	Broad Institute
		*Myotis brandtii* (Brandt’s bat)/GCF_000412655.1	2.11	1981.77	169,750 Scaffold	3225.83	BGI
		*Lasiurus borealis* (red bat)/GCA_004026805.1	2.86	0.64	518,900 Scaffold	38.54	Broad Institute
		*Murina aurata feae* (little tube-nosed bat)/GCA_004026665.1	2.33	0.68	880,177 Scaffold	26.05	Broad Institute
		*Nycticeius humeralis* (evening bat)/GCA_007922795.1	2.78	0.43	1,676,240 Scaffold	15.09	Broad Institute
		*Pipistrellus pipistrellus* (common pipistrelle)/GCA_903992545.1	1.76	0.47	22 Chromosome	94,929.99	SC
		*Myotis myotis* (greater mouse-eared bat)/GCA_014108235.1	2	28.95	92 Scaffold	94,448.91	Bat1K
		*Myotis yumanensis* (Yuma Myotis)/GCA_028538775.1	1.95	0.02	475 Scaffold	99,144.70	UCLA
		*Pipistrellus kuhlii* (Kuhl’s pipistrelle)/GCA_014108245.1	1.78	12.2	202 Scaffold	80,237.35	Bat1K
		*Myotis davidii* (David’s Myotis)/GCA_000327345.1	2.06	181.34	101,769 Scaffold	3454.48	BGI
	Rhinolophidae	*Rhinolophus ferrumequinum* (greater horseshoe bat)/GCA_004115265.3	2.08	7.55	28 Chromosome	88,025.74	Vertebrate Genomes Project
	Phyllostomidae	*Phyllostomus discolor* (pale spear-nosed bat)/GCA_004126475.3	2.11	15.09	17 Chromosome	171,742.86	Vertebrate Genomes Project
		*Desmodus rotundus* (common vampire bat)/GCA_022682495.1	2.11	0.26	14 Chromosome	160,127.86	Bat1K
		*Phyllostomus hastatus* (greater spear-nosed bat)/GCA_019186645.2	2.09	0.01	534 Scaffold	39,158.28	Texas Tech University
		*Artibeus jamaicensis* (Jamaican fruit-eating bat)/GCA_021234435.1	2.11	0	867 Scaffold	22,092.64	CSHL
		*Anoura caudifer* (tailed tailless bat)/GCA_004027475.1	2.21	0.46	337,255 Scaffold	185.02	Broad Institute
		*Tonatia saurophila* (stripe-headed round-eared bat)/GCA_004024845.1	2.11	0.29	249,810 Scaffold	165.56	Broad Institute
		*Macrotus californicus* (California big-eared bat)/GCA_007922815.1	2.16	0.21	1,125,430 Scaffold	16.94	Broad Institute
		*Carollia perspicillata* (Seba’s short-tailed bat)/GCA_004027735.1	2.69	0.39	1,925,339 Scaffold	10.73	Broad Institute
		*Trachops cirrhosus* (fringe-lipped bat)/GCA_028533065.1	2.18	14.91	15 Chromosome	124,458.21	DNA Zoo
		*Micronycteris hirsuta* (hairy big-eared bat)/GCA_004026765.1	2.31	0.32	550,090 Scaffold	68.87	Broad Institute
		Sturnira hondurensis (bats)/GCA_014824575.3	2.1	4.31	25,881 Scaffold	10,164.81	College of Life Sciences at Wuhan University
	Noctilionidae	*Noctilio leporinus* (greater bulldog bat)/GCA_004026585.1	2.1	0.62	298,222 Scaffold	191.49	Broad Institute
	Mormoopidae	*Pteronotus parnellii mesoamericanus* (Parnell’s mustached bat)/GCA_021234165.1	2.07	0	333 Scaffold	31,477.94	CSHL
		*Mormoops blainvillei* (Antillean ghost-faced bat)/GCA_004026545.1	2.11	0.18	205,259 Scaffold	156.29	Broad Institute
	Molossidae	*Molossus molossus* (Pallas’s mastiff bat)/GCA_014108415.1	2.32	47.37	60 Scaffold	110,665.20	Bat1K
		*Molossus nigricans* (northern black mastiff bat)/GCA_026936385.1	2.41	0.1	146 Scaffold	81,933.98	Bat1K
		*Tadarida brasiliensis* (Brazilian free-tailed bat)/GCA_004025005.1	2.71	0.42	1,067,615 Scaffold	24.31	Broad Institute
	Miniopteridae	*Miniopterus schreibersii* (Schreibers’ long-fingered bat)/GCA_004026525.1	1.78	0.69	177,620 Scaffold	108.71	Broad Institute
		*Miniopterus natalensis* (Natal long-fingered bat)/GCA_001595765.1	1.8	68.17	1269 Scaffold	4315.19	University of California, San Francisco
	Megadermatidae	*Megaderma lyra* (Indian false vampire)/GCA_004026885.1	2.62	0.63	1,902,801 Scaffold	96.49	Broad Institute
	Hipposideridae	*Hipposideros pendleburyi* (Pendlebury’s leaf-nosed bat)/GCA_021464545.1	2.17	44.77	28,685 Scaffold	15,398.51	National Science and Technology Development Agency
		*Hipposideros armiger* (great roundleaf bat)/GCF_001890085.2	2.24	1954.59	7386 Scaffold	2328.17	SKLEC and IECR
		*Hipposideros galeritus* (Cantor’s roundleaf bat)/GCA_004027415.1	2.44	0.64	840,200 Scaffold	37.98	Broad Institute
	Craseonycteridae	*Craseonycteris thonglongyai* (hog-nosed bat)/GCA_004027555.1	2.27	0.62	1,224,256 Scaffold	25.76	Broad Institute
Megachiroptera	Pteropodidae	*Eonycteris spelaea* (lesser dawn bat)/GCA_003508835.1	1.97	0	4469 Contig	8002.59	Duke–NUS Medical School
		*Cynopterus brachyotis* (lesser short-nosed fruit bat)/GCA_009793145.1	1.76	71.77	48,006 Scaffold	251.27	National University of Singapore
		*Cynopterus sphinx* (Indian short-nosed fruit bat)/GCA_030015415.1	1.9	-	17 Chromosomes	145.2	Wuhan University
		*Eidolon helvum* (straw-colored fruit bat)/GCA_000465285.1	1.84	7.32	133,538 Scaffold	27.68	School of Biological and Chemical Sciences, Queen Mary University of London
		*Eidolon dupreanum* (Malagasy straw-colored fruit bat)/GCA_028627145.1 (latest)	2.29	8.09	17 Chromosome	101,563.12	DNA Zoo
		*Pteropus rufus* (Malagasy flying fox)/GCA_028533765.1	2.09	4.65	19 Chromosome	110,476.78	DNA Zoo
		*Rousettus madagascariensis* (Madagascan rousette)/GCA_028533395.1	2.34	10.35	18 Chromosome	85,834.86	DNA Zoo
		*Rousettus aegyptiacus* (Egyptian rousette)/GCA_014176215.1	1.89	26.67	29 Scaffold	113,811.80	Bat1K
		*Pteropus alecto* (black flying fox)/GCA_000325575.1	1.99	41.33	65,597 Scaffold	15,954.80	BGI
		*Pteropus vampyrus* (large flying fox)/GCA_000151845.2	2.2	181.04	36,094 Scaffold	5954.02	Baylor College of Medicine
		*Macroglossus sobrinus* (lesser long-tongued fruit bat)/GCA_004027375.1	1.9	0.21	171,985 Scaffold	453.40	Broad Institute
		*Pteropus giganteus* (Indian flying fox)/GCA_902729225.1	1.99	17.91	16,113 Scaffold	18,871.25	CIRI-Inserm-U1111
		*Rousettus leschenaultii* (Leschenault’s rousette)/GCA_015472975.1	1.92	16.88	8141 Scaffold	32,720.14	Comparative Genomics Laboratory, Center for Information Biology, National Institute of Genetics
		*Pteropus pselaphon* (Bonin flying fox)/GCA_014363405.1	1.93	0.79	7513 Scaffold	770.41	Environmental Genomics Office, Center for Environmental Biology and Ecosystem Studies, National Institute for Environmental Studies

## Data Availability

All data are available in the main text.

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
