# Peer review of "Different Species of Bats: Genomics, Transcriptome, and Immune Repertoire"

_cimb, 2025, doi:10.3390/cimb47040252_

Round 1
Reviewer 1 Report
Comments and Suggestions for Authors
In this study, authors comparative analysis the differences in genome size, composition, and expression levels of genes related to hearing, vision, and olfaction, and immune gene composition features among different bat species. The review tries to reveal the underlying mechanisms enabling bats to harbor diverse pathogenic viruses while remaining asymptomatic, with the goal of informing clinical applications. It's a intesting paper. However, the abstract should be revised to incorporate these variations, as their omission may compromise the transparency of the findings. Moreover,The future research directions must be explicitly derived from the current findings to guide subsequent investigations, instead of reiterating generic statements. A rigorously constructed perspectives section is pivotal in contextualizing this study’s contributions to the field. The manuscript needs careful editing throughout the manuscript before it may be considered of publication quality.
Author Response
Reviewer 1
In this study, authors comparative analysis the differences in genome size, composition, and expression levels of genes related to hearing, vision, and olfaction, and immune gene composition features among different bat species. The review tries to reveal the underlying mechanisms enabling bats to harbor diverse pathogenic viruses while remaining asymptomatic, with the goal of informing clinical applications. It's a intesting paper. However, the abstract should be revised to incorporate these variations, as their omission may compromise the transparency of the findings. Moreover,The future research directions must be explicitly derived from the current findings to guide subsequent investigations, instead of reiterating generic statements. A rigorously constructed perspectives section is pivotal in contextualizing this study’s contributions to the field. The manuscript needs careful editing throughout the manuscript before it may be considered of publication quality.
Thank you very much for your detailed comments and valuable suggestions.
- We have revised the Abstract to address genomic and sensory gene variations across bat species, ensuring alignment with the manuscript’s core findings. Key additions include:
In R1 Line 15:we made the change “Bats are the only mammals with the ability to fly and are the second largest order after rodents, with 20 families and 1,213 species (over 3,000 subspecies) and are widely distributed in regions around the world except for Antarctica. What makes bats unique are their biological traits: tolerance to zoonotic infections without getting clinical symptoms, long lifespans, a low incidence of tumors, and high metabolism. As a result, they are receiving increasing attention in the field of life sciences, particularly in medical research.The rapid advancements in sequencing technology have made it feasible to comprehensively analyze the diverse biological characteristics of bats. This review comprehensively discusses: (1) the assembly and annotation overview of 77 assemblies from 54 species across 11 families, and the transcriptome sequencing overview of 42 species from 7 families,focused on a comparative analysis of genomic architecture, sensory adaptations (auditory, visual, and olfactory), and immune functions. Key findings encompass marked interspecies divergence in genome size, lineage-specific expansions/contractions of immune-related gene families (APOBEC, IFN, PYHIN), and sensory gene adaptations linked to ecological niches. Notably, echolocating bats exhibited convergent evolution in auditory genes (SLC26A5, FOXP2), while fruit-eating bats displayed degeneration of vision-associated genes (RHO), reflecting trade-offs between sensory specialization and ecological demands.(2) the annotation of V (variable), D (diversity), J (joining), and C (constant) gene families in the TR and IG loci of 12 species from 5 families, with a focus on comparative analysis of the differences in TR and IG genes and CDR3 repertoires between different bats and between bats and other mammals,provides us with a deeper understanding of the development and function of the immune system in organisms.Integrated genomic, transcriptomic, and immune repertoire analyses reveal that bats employ distinct antiviral strategies, primarily mediated by enhanced immune tolerance and suppressed inflammatory responses This review provides foundational information, collaboration directions, and new perspectives for various laboratories conducting basic and applied research on the vast array of bat biology.”
- We agree that the original "Future Directions" section lacked specificity. In R1 Section 4 ("Summary and Outlook"), we now propose targeted research priorities directly tied to our findings:
In R1 Line 447:we made the change “Analyzing the genome, transcriptome and immune repertoire of different bats can clarify their mechanisms and tolerance strategies as hosts of various viruses, and provide new ideas and methods for research. With the development of sequencing and annotation of bat genome has promoted the development of new methods for studying virus tolerance mechanism and provided technical support for clarifying its mechanism of action,such as recent genomic analyses of bats have revealed diverse antiviral adaptations, including expansions of APOBEC3 loci, positive selection in interferon-stimulated genes (ISGs), and lineage-specific losses of pro-inflammatory receptors, suggesting that antiviral mechanisms in bats evolved through multiple independent pathway[Morales, A.E.; Dong, Y.; Brown, T.; Baid, K.; Kontopoulos, D.-G.; Gonzalez, V.; Huang, Z.; Ahmed, A.-W.; Bhuinya, A.; Hilgers, L.; et al. Bat Genomes Illuminate Adaptations to Viral Tolerance and Disease Resistance. Nature 2025, 638, 449–458, doi:10.1038/s41586-024-08471-0.]. In-depth analysis of bat transcriptome enables us to analyze the dynamic expression rules of its immune-related genes, such as constitutively expressing interferon-α (IFN-α) in bats to maintain antiviral state[24]. Notably, the multi - level gene expression regulation synergizes with the highly diverse BCR/TCR repertoire. The latter, generating broad - spectrum antigen - recognition receptors, enables precise pathogen attack. These studies may represent the main strategy of bats to coexist with viruses.
The establishment of bat cell model provides an important tool for studying bat's unique antiviral ability.For instance, Crameri G. et al.[81] established immortalized cell lines of Pteropus alecto in 2009 to examine its susceptibility to rabies and Hendra viruses, among others.”
- We have revised and polished the English and grammar of the full text for readers' better understanding. For instance,
In R1 Line 38:we made the change “Integrated genomic, transcriptomic, and immune repertoire analyses reveal that bats employ distinct antiviral strategies, primarily mediated by enhanced immune tolerance and suppressed inflammatory responses.”
In R1 Line 50:we made the change “In recent years, sequencing technologies have undergone transformative progress, transitioning from single-fragment Sanger sequencing to next-generation sequencing (NGS) platforms (e.g., Illumina short-read systems), and more recently to long-read sequencing technologies (e.g., PacBio SMRT and Oxford Nanopore).”
In R1 Line 55:we made the change “Bats, capable of hosting numerous viruses without exhibiting overt disease symptoms, have drawn attention to their immune systems.”
In R1 Line 58:we made the change “However, due to the vast diversity of Chiroptera with 20 families, 1,213 species, and over 3,000 subspecies,it is a very difficult task to comprehensively sequence the genome of bat[5]”
We sincerely appreciate the reviewer’s meticulous feedback, which has significantly strengthened the clarity, specificity, and impact of our work. All changes are highlighted in the revised manuscript for ease of evaluation.

Reviewer 2 Report
Comments and Suggestions for Authors
The manuscript reviews the important and timely topic of bat genomics and adaptation.
In terms of adaptation in the immune system, the authors keep mentioning “tolerance” while disregarding “resistance” mechanism. They should explicitly define the two terms and then provide several examples of each of these two mechanisms in different bat species to demonstrate the differences between them. Several examples for these appear, especially in regards to fruit bats (but not only) in several papers and reviews (e.g., PMID: 26805873, 37575178, 34010652, 33157026, 36851566, 39836373, 33147460,38453576, 38193073, 32117225, 28959255,)
Figure 1 – It would be better to put the arrows of “sequenced” and “not sequenced” in the same group of bat species and not in two (and to use “not” rather than “non”).
Technical comments:
In the Abstract:
There are three sentences without a space after the point:
"As a result, they are receiving increasing attention in the field of
life sciences, particularly in medical research.The disruptive advancements in sequencing technology have made it feasible to comprehensively analyze the diverse biological characteristics of bats.This review comprehensively discusses:"
There are additional such sentences in the Abstract.
Also the term "disruptive" does not make sense in the context of the above sentence.
In summary:
…”such as the recent discovery of evidence of "synergistic" evolution of multiple viral genes in a large number of bat genome sequences(Hiller et al., 2023).”
- The preprint can now be replaced with the published paper
- The term should be “antiviral genes” and not “viral genes”
- There is no direct evidence for synergistic evolution in the mentioned paper, but rather that there are multiple pathways in which antiviral adaptations may have occurred
Comments on the Quality of English Language
The manuscript could be improved by having a native English speaker, going over it and rephrasing some of the sentences.
Author Response
Reviewer 2
Comments:In terms of adaptation in the immune system, the authors keep mentioning “tolerance” while disregarding “resistance” mechanism. They should explicitly define the two terms and then provide several examples of each of these two mechanisms in different bat species to demonstrate the differences between them. Several examples for these appear, especially in regards to fruit bats (but not only) in several papers and reviews (e.g., PMID: 26805873, 37575178, 34010652, 33157026, 36851566, 39836373, 33147460,38453576, 38193073, 32117225, 28959255,)
Response:Thank you very much for your detailed comments and valuable suggestions.
We sincerely appreciate the reviewer’s insightful feedback. Below, we clarify the definitions of “resistance” and “tolerance” and provide examples of both mechanisms across bat species, incorporating key references as suggested.
In R1 Line 106: we made the change “Gene expansion and contraction within the bat genome suggest that bats may have evolved distinct strategies to respond antigen invasions.Emerging evidence suggests that bat immune systems balance viral coexistence and host survival through dual strategies: immune tolerance and immune resistance. Notably, some bats maintain rapid antiviral defenses via constitutively active innate immunity. For example, comparative studies in Rousettus aegyptiacus and Myotis myotis show both species depend on interferon signaling, yet diverge in pathway utilization,Rousettus aegyptiacus prioritizes TLR3 activation, while Myotis myotis relies on the RIG-I pathway,highlighting evolutionary flexibility in antiviral tactics[Schneor, L.; Kaltenbach, S.; Friedman, S.; Tussia-Cohen, D.; Nissan, Y.; Shuler, G.; Fraimovitch, E.; Kolodziejczyk, A.A.; Weinberg, M.; Donati, G.; et al. Comparison of Antiviral Responses in Two Bat Species Reveals Conserved and Divergent Innate Immune Pathways. iScience 2023, 26, 107435, doi:10.1016/j.isci.2023.107435.].Levinger R et al. (2025) used single cell and spatial transcriptome technology to reveal the unique distribution of immune cells (such as high proportion of regulatory T cells) in bat skin and respiratory barrier tissues and the low expression of virus receptors (such as ACE2) in epithelial cells, limiting virus invasion[Levinger, R.; Tussia-Cohen, D.; Friedman, S.; Lender, Y.; Nissan, Y.; Fraimovitch, E.; Gavriel, Y.; Tearle, J.L.E.; Kolodziejczyk, A.A.; Moon, K.-M.; et al. Single-Cell and Spatial Transcriptomics Illuminate Bat Immunity and Barrier Tissue Evolution. Mol Biol Evol 2025, 42, msaf017, doi:10.1093/molbev/msaf017.]. Other studies have shown that there are differences in interferon type I among different species of bats,such as Pavlovich et al.[24] discovered positive selection in Rousettus aegyptiacus for interferon-stimulated genes ISG15, type I IFN receptor subunit IFNAR1, and interferon response negative regulator SIKE1 in Rousettus aegyptiacus. The IFN-ω gene has expanded extensively to nearly 20 copies in this species, a contrast to the expansion observed in Pteropus alecto but is consistent with Pteropus vampyrus.
In the tolerance mechanism,bats can achieve asymptomatic carrying of highly pathogenic viruses (such as Ebola virus and Marburg virus) by regulating inflammatory response and metabolic adaptation, including IRF1 inhibits excessive inflammation by up-regulating immune checkpoint molecules (such as PD-L1)[Irving, A.T.; Zhang, Q.; Kong, P.-S.; Luko, K.; Rozario, P.; Wen, M.; Zhu, F.; Zhou, P.; Ng, J.H.J.; Sobota, R.M.; et al. Interferon Regulatory Factors IRF1 and IRF7 Directly Regulate Gene Expression in Bats in Response to Viral Infection. Cell Rep 2020, 33, 108345, doi:10.1016/j.celrep.2020.108345.],inhibit the expression of proinflammatory factors (IL-6, TNF-α, IL-1β, IL-8) to alleviate tissue damage[Paweska, J.T.; Storm, N.; Grobbelaar, A.A.; Markotter, W.; Kemp, A.; Jansen van Vuren, P. Experimental Inoculation of Egyptian Fruit Bats (Rousettus Aegyptiacus) with Ebola Virus. Viruses 2016, 8, 29, doi:10.3390/v8020029.][Jayaprakash, A.D.; Ronk, A.J.; Prasad, A.N.; Covington, M.F.; Stein, K.R.; Schwarz, T.M.; Hekmaty, S.; Fenton, K.A.; Geisbert, T.W.; Basler, C.F.; et al. Marburg and Ebola Virus Infections Elicit a Complex, Muted Inflammatory State in Bats. Viruses 2023, 15, 350, doi:10.3390/v15020350.],high-frequency expression of IGHV1 family genes in B cells is used to produce broad-spectrum antibodies, and low-inflammatory antibody subtypes such as IgG4 are relied on to reduce immune activation[Larson, P.A.; Bartlett, M.L.; Garcia, K.; Chitty, J.; Balkema-Buschmann, A.; Towner, J.; Kugelman, J.; Palacios, G.; Sanchez-Lockhart, M. Genomic Features of Humoral Immunity Support Tolerance Model in Egyptian Rousette Bats. Cell Rep 2021, 35, 109140, doi:10.1016/j.celrep.2021.109140.],the up-regulation of T cell immune checkpoint (PD-1/CTLA-4) inhibits overreaction, and enhances antioxidant defense (such as the high expression of SOD2 and the deletion of PYHIN family sites) to alleviate virus-induced oxidative stress[Guito, J.C.; Prescott, J.B.; Arnold, C.E.; Amman, B.R.; Schuh, A.J.; Spengler, J.R.; Sealy, T.K.; Harmon, J.R.; Coleman-McCray, J.D.; Kulcsar, K.A.; et al. Asymptomatic Infection of Marburg Virus Reservoir Bats Is Explained by a Strategy of Immunoprotective Disease Tolerance. Curr Biol 2021, 31, 257-270.e5, doi:10.1016/j.cub.2020.10.015.][30][31],so as to maintain the host-pathogen balance under the persistent existence of virus. In addition, Vicente-Santos A et al. found that the serum of Desmodus rotundus highly expressed complement inhibitory protein (such as CFH) and anti-inflammatory factor (such as IL-10), which supported wide tolerance to many pathogens[Vicente-Santos, A.; Lock, L.R.; Allira, M.; Dyer, K.E.; Dunsmore, A.; Tu, W.; Volokhov, D.V.; Herrera, C.; Lei, G.-S.; Relich, R.F.; et al. Serum Proteomics Reveals a Tolerant Immune Phenotype across Multiple Pathogen Taxa in Wild Vampire Bats. Front Immunol 2023, 14, 1281732, doi:10.3389/fimmu.2023.1281732.].”
Comments:Figure 1 – It would be better to put the arrows of “sequenced” and “not sequenced” in the same group of bat species and not in two (and to use “not” rather than “non”).
Response:Thank you very much for your detailed comments and valuable suggestions.
We have revised Figure 1 by placing the "sequenced" and "not sequenced" arrows within the same bat species group and changed "non-sequenced" to "not sequenced". Thank you for your valuable suggestions.
Comments:Technical comments:
In the Abstract:
There are three sentences without a space after the point:
"As a result, they are receiving increasing attention in the field of life sciences, particularly in medical research.The disruptive advancements in sequencing technology have made it feasible to comprehensively analyze the diverse biological characteristics of bats.This review comprehensively discusses:"
There are additional such sentences in the Abstract.Also the term "disruptive" does not make sense in the context of the above sentence.
Response:Thank you very much for your detailed comments and valuable suggestions.
In R1 Line 23:we made the change “ This review comprehensively discusses: (1) the assembly and annotation overview of 77 assemblies from 54 species across 11 families, and the transcriptome sequencing overview of 42 species from 7 families,focused on a comparative analysis of genomic architecture, sensory adaptations (auditory, visual, and olfactory), and immune functions. Key findings encompass marked interspecies divergence in genome size, lineage-specific expansions/contractions of immune-related gene families (APOBEC, IFN, PYHIN), and sensory gene adaptations linked to ecological niches. Notably, echolocating bats exhibited convergent evolution in auditory genes (SLC26A5, FOXP2), while fruit-eating bats displayed degeneration of vision-associated genes (RHO), reflecting trade-offs between sensory specialization and ecological demands.(2) the annotation of V (variable), D (diversity), J (joining), and C (constant) gene families in the TR and IG loci of 12 species from 5 families, with a focus on comparative analysis of the differences in TR and IG genes and CDR3 repertoires between different bats and between bats and other mammals,provides us with a deeper understanding of the development and function of the immune system in organisms.Integrated genomic, transcriptomic, and immune repertoire analyses reveal that bats employ distinct antiviral strategies, primarily mediated by enhanced immune tolerance and suppressed inflammatory responses This review provides foundational information, collaboration directions, and new perspectives for various laboratories conducting basic and applied research on the vast array of bat biology.”
In R1 Line 21:we made the change “The disruptive advancements in sequencing technology have made it feasible to comprehensively analyze” to “The rapid advancements in sequencing technology have made it feasible to comprehensively analyze”
Comments:
In summary:
…”such as the recent discovery of evidence of "synergistic" evolution of multiple viral genes in a large number of bat genome sequences(Hiller et al., 2023).”
- The preprint can now be replaced with the published paper
- The term should be “antiviral genes” and not “viral genes”
- There is no direct evidence for synergistic evolution in the mentioned paper, but rather that there are multiple pathways in which antiviral adaptations may have occurred
Response:Thank you very much for your detailed comments and valuable suggestions. We updated the references and re-described the contents:
In R1 Line 452:we made the change “such as recent genomic analyses of bats have revealed diverse antiviral adaptations, including expansions of APOBEC3, positive selection in interferon-stimulated genes (ISGs), and lineage-specific losses of pro-inflammatory receptors, suggesting that antiviral mechanisms in bats evolved through multiple independent pathways[Morales AE, Dong Y, Brown T, Baid K, Kontopoulos D-, Gonzalez V, Huang Z, Ahmed AW, Bhuinya A, Hilgers L, Winkler S, Hughes G, Li X, Lu P, Yang Y, Kirilenko BM, Devanna P, Lama TM, Nissan Y, Pippel M, Dávalos LM, Vernes SC, Puechmaille SJ, Rossiter SJ, Yovel Y, Prescott JB, Kurth A, Ray DA, Lim BK, Myers E, Teeling EC, Banerjee A, Irving AT, Hiller M. Bat genomes illuminate adaptations to viral tolerance and disease resistance. Nature. 2025 Feb;638(8050):449-458.]”
Comments:The manuscript could be improved by having a native English speaker, going over it and rephrasing some of the sentences.
Response:Thank you very much for your detailed comments and valuable suggestions. We have polished and revised the English and grammar of the full text.
In R1 Line 38:we made the change “Integrated genomic, transcriptomic, and immune repertoire analyses reveal that bats employ distinct antiviral strategies, primarily mediated by enhanced immune tolerance and suppressed inflammatory responses.”
In R1 Line 50:we made the change “In recent years, sequencing technologies have undergone transformative progress, transitioning from single-fragment Sanger sequencing to next-generation sequencing (NGS) platforms (e.g., Illumina short-read systems), and more recently to long-read sequencing technologies (e.g., PacBio SMRT and Oxford Nanopore).”
In R1 Line 55:we made the change “Bats, capable of hosting numerous viruses without exhibiting overt disease symptoms, have drawn attention to their immune systems.”
In R1 Line 58:we made the change “However, due to the vast diversity of Chiroptera with 20 families, 1,213 species, and over 3,000 subspecies,it is a very difficult task to comprehensively sequence the genome of bat[5]”
In R1 Line 235:we made the change “Notably, transcriptomes of Rousettus aegyptiacus uncovered constitutive expression of antiviral effectors (IFN-γ, IFN-α) and pattern recognition receptors, yet strikingly lacked baseline IFN-β expression,consistent with other mammals in which IFNB is expressed only after stimulation.”

Round 2
Reviewer 1 Report
Comments and Suggestions for Authors
The authors have significantly improved their manuscript. They well-addressed all of my comments. The paper can be accepted for publication after the editorial check.